# The BernClim plant phenological data set from the Canton of Bern (Switzerland) 1970–2018

This Rutishauser[1,2], François Jeanneret[2], Robert Brügger[3], Yuri Brugnara[1,2], Christian Röthlisberger[4], August Bernasconi[5], Peter Bangerter[6], Céline Portenier[1,2], Leonie Villiger[1,2], Daria Lehmann, Lukas Meyer[1,2], Bruno Messerli[2†], Stefan Brönnimann[1,2]

[1] Oeschger Centre for Climate Change Research, University of Bern, Berne, 3012, Switzerland
[2] Institute of Geography, University of Bern, Berne, 3012, Switzerland
[3] Grossaffoltern, 3257, Switzerland
[4] Langnau, 3550, Switzerland
[5] Einigen, 3646, Switzerland
[†] deceased

Correspondence to: This Rutishauser (this@kontextlabor.ch)

**Abstract.** In 1970, the Institute of Geography of the University of Berne initiated the phenological observation network BernClim. Seasonality information from plants, fog and snow originally served for applications in urban and regional planning, agricultural and touristic suitability and are now a valuable data set for climate change impacts studies. Covering the growing season volunteer observers record the dates of key development stages of hazel (Coryllus avellana), dandelion (Taraxacum officinale), apple tree (Pyrus malus) and beech (Fagus sylvatica). All observations consist of detailed site information including location, altitude, exposition (aspect) and inclination that make BernClim unique in detail-richness on decadal time-scales. Quality control (QC) by experts and statistical analyses of the data has been performed to flag impossible dates, dates outside the biologically plausible range, repeated dates in the same year, stretches of consecutive identical dates, and statistically inconsistent dates (outliers in time or in space). Here, we report BernClim data of 7414 plant phenological observations from 1970 to 2018 from 1304 sites at 110 stations, the QC procedure and selected applications (Rutishauser et al. 2019, doi: https://doi.org/10.1594/PANGAEA.900102). The QC points to a very good internal consistency (only 0.2% were flagged as internally inconsistent) and likely a high quality of the data. BernClim data indicate a trend towards an extended growing season. They also well track the regime shift in the late 1980s to pronounced earlier dates like numerous other phenological records across the Northern Hemisphere.

**Keywords:** phenology, phenological observations, cherry, apple, dandelion, hazel, beech, snow, fog

## 1 Introduction

The seasonality of plants has been observed for centuries for tracking and documenting plant agricultural practices (Schnelle 1955, Demarée and Rutishauser 2011, Rutishauser 2009, Pfister and White 2018, Jeanneret et al. 2018). Systematic documentations started with the famous Kyoto cherry series as early as 801 AD (Aono and Kazui 2008). Phenological phases indicate growth cycle stages of annual and perennial plant life. The stages are closely linked to environmental drivers such as light, temperature and precipitation and are, thus, used as climate change impact indicators (Menzel et al 2006, IPCC 2013) or serve as proxy data in climate reconstructions (Rutishauser et al. 2008; Anderson et al. 2013, Ge et al., 2014). For instance, the inventory of climate monitoring series of the Swiss GCOS Office (MeteoSwiss 2018) lists phenological records. Peñuelas et al (2008) stressed the importance of phenological observations for climate feedback processes that can only be studied when ground observational data are available in a reasonable quality (Rutishauser et al. 2009).

In Switzerland, the longest continuous phenologial series reach back to 1808 (Horse chestnut bud burst, Geneva) and 1894 (Cherry flowering, Liestal; Defila and Clot 2001, Defila et al. 2016). In Europe, single observations are documented from the High Middle Ages onwards (Pfister and White 2018). Systematic collections started in the 18th century, e.g., several decades from 1760 onwards by the «Ökonomische Gesellschaft Bern» (Pfister 1999), and from 1869 to 1882 by the Forestry Department of the Canton of Bern (Vassella 1997). In 1951, the Swiss Phenology Network (SPN) has been initiated (Primault et al. 1957; Defila and Clot 2001; Studer et al. 2005; MeteoSwiss 2018). Today, SPN comprises 160 stations, distributed across various regions and elevations of Switzerland. Each year, observers record the dates of leaf unfolding (needle appearance), flowering, fruit ripening, leaf colouring and leaf fall for selected wild plants and crops. These observations cover 26 plant species and 69 phenophases (MeteoSwiss 2018, Auchmann et al. 2018).

In 1969 and complementing the SPN, the BernClim phenological network was established by the late Bruno Messerli of the Institute of Geography, University of Bern (Messerli et al. 1978, Jeanneret and Rutishauser 2012). The aim was to provide a scientific basis for complex climate studies and spatial planning, specifically for determining agricultural and touristic suitability and assessing natural hazards. At higher spatial resolution and precision, the BernClim network systematically documented specific coordinates of observation sites, exposition (aspect) and inclination that were combined to stations.

Quality control and assurance of phenological series have become increasingly important for newly generated data as well as archive observations. In Switzerland, efforts have been undertaken in a recent Swiss GCOS project (Auchmann et al. 2018). Data sets have been compiled for Europe within the Pan European Phenology Project PEP725 (www.pep725.eu, Menzel et al., 2006), and for the USA (e.g., Rosemartin et al. 2015). In addition, comparative analyses from networks and Citizen Science Projects suggest that different data sources are complementary depending on the research question. Most recent analyses showed that observations from Citizen Science Projects PhaenoNet and OpenNature complement the data from the professional network SPN qualitywise (Lehmann et al. 2018). Differences can be explained by the extent and uneven distribution of the spatial coverage. Near-realtime visualizations and comparisons can now be combined with archived observations back to 1951.

In this paper we describe the plant phenological observations and quality control efforts of the BernClim data set for phenological and climatological analyses publicly available from PANGEAE (Rutishauser et al., 2019) and from the PEP-725 database, which soon spans half a century. In Section 2, we provide background on the observation network and give an overview of the data. Section 3 describes the results of the quality control. In Section 4 we then present selected results and draw conclusions in Section 5.

## 2 Observation Network and Data

The BernClim observation network focuses on the territory of the Canton of Bern (Switzerland, Figure 1). The Canton of Bern stretches across three major Swiss landscapes from the Jura mountains across Swiss Plateau to the Alps and spans an altitudinal range from 400 to 4000 m a.s.l.. Climate in the study region is determined by westerly, northwesterly and southwesterly winds (i.e., from the Atlantic Ocean) and the passage of weather systems. In summer, the Azores high is the dominant pressure system, alternating with westerly and northerly flow situations. Regional wind systems such as Föhn and the Bise may play an important role. Most areas receive an adequate amount of precipitation throughout the year.

BernClim was initiated as a five-year research project funded by the Canton of Bern in 1969 and grew into a still ongoing observation programme coordinated by the University of Bern (Messerli et al. 1978, Jeanneret and Rutishauser 2012). Observations began in 1970. The main observation phase of the project lasted from 1970 to 1974 with the final report compiled by Messerli et al. (1978). A detailed overview, such as on how observers were located, trained and details on observation guidelines and site selection are given in Jeanneret & Rutishauser 2012.

Following the success of the first phase, the project was continued with funding from diverse sources. Many observers continued and the network has been operated ever since by the Institute of Geography of the University of Bern, during the last 3 decades as a non-funded activity. Apart from serving spatial planning, the BernClim data have been used in education. While the number of observers has steadily decreased, five have remained to the present day. These long term series are today a valuable source of information also for science, particularly as there were only few observer changes throughout the network.

To cover all four seasons, observation periods were divided into growing and resting periods. Plant phenology from early spring to late autumn documents summers. During winter, fog presence and duration and snow cover were observed daily from late autumn to early spring (Table 1).

The definition of the plant phenological observations follows the official instructions of MeteoSwiss (Jeanneret 1971, Primault 1957, Brügger and Vassella 2018). Overall, more than 200 volunteers were recruited for observing in 1971 through the teacher training program of the Institute of Geography. A large number of observers have a training in public school teaching or family doctors, and have a strong, intrinsic motivation for observing natural phenomena and processes. Data was submitted from 180 stations in 1971 with station and site numbers decreasing since (Figure 2). The spatial representativity of stations strongly reflects population density. All volunteers were asked to select a number of locally representative sites (in the following "observation sites") mostly in cultivated systems. A comprehensive overview of the BernClim network was published in Jeanneret and Rutishauser (2012).

Phenological phases are defined by a morphological development phase of a plant that has to be reached as well as a quantitative threshold. The observation then is the date (day of year, DoY) when this threshold is crossed. For instance, for the case of apple trees, general flowering is reached when 50% of the blossoms are „open". The definition of „open" is morphologically described in the observers instructions. Each plant and phenological phase was noted on a specific form (Figure 3).

## 2.2 Data

In this paper we describe 7414 quality controlled plant phenological observations from 1970 to 2018 (Rutishauser et al. 2019). Data were collected at sites between altitudes from 410 and 1700 m a.s.l. Reported plant species and phenological phases include the flowering of hazel (*Coryllus avellana*), dandelion (*Taraxacum officinale*) and apple trees (*Pyrus malus*), the leave colouring of beech (*Fagus sylvatica*). Each observation record contains the site information including a popular site name (toponym), coordinates, altitude, exposition and inclination. Several sites are combined to stations that are labeled with zip codes.

The different phases of the network yield quite different numbers of observations. During the intensive initial phase of the network, around 123.500 data were collected. A large number of observations were single observations and were not quality checked for this study. The number of stations and observation sites gradually decreased from initially 76 and 448, respectively. Presently there are 5 stations and observers. Figure 2 shows the number of stations as time series. Although the number of stations decreased rapidly, even the current, very sparse network still has each of the three major landscapes represented.

## 3. Raw Data and Quality Control

The observers received standard forms to fill out and send back by regular mail. Figure 3 shows an example of a data sheet for plant phenophases. Figures 4 and 5 show the form used for snow and fog, respectively. All original observation sheets of plant, snow and fog observations are archived at the University of Bern. During ongoing data rescue a large fraction has been photographed. To date, only plant observations have been digitized and controlled for publication.

The quality control (QC) process consisted of several steps. First, the raw data were read into a GIS for coordinate checking. Only wrong coordinates, altitudes or location names were corrected (see Kottmann 2008, for details). Except for very obvious errors, which were deleted, the observed dates were not changed.

The second step consisted of an operational baseline QC, which was done by an expert in biology, plant physiology and phenology (Robert Brügger). This step included filling of data gaps from the original paper records and station history descriptions. During this procedure, observer changes were systematically recorded in station documentations including interviews with observers (unpublished data).

The third step comprised an automated flagging routine similar to Auchmann et al (2018). This automatic quality control of the BernClim data consisted of six checks. For this purpose we formed „series", which refers to all events of the same

phenological phase at the same site (i.e., same coordinates). This means that there can be a large number of series per station and zip code. The first four tests use absolute dates, test five is based on standardised series while for test six, for a given year, the standardised dates from all series were re-standardized. The following flags were set:

Test 1: impossible dates (day of year above 366, below -366, or 0 are considered impossible)

         Test 2: dates outside of the range indicated by MeteoSwiss (personal communication, Tab. 2)

         Test 3: non-first dates (if several dates are found in the same year, all except the first were flagged)

         Test 4: four consecutive identical dates after removing non-first dates of the same year

         Test 5: dates outside of ±3 standard deviations (sd) of each series after removing non-first dates of the same year and

only for series with a minimum length of 10 (41% of all values tested)

         Test 6: dates outside of ±3 standard deviations of all series for a given year after removing non-first dates of the same

         year and only for series with a minimum length of 10 and years with a minimum of 10 observations (40% of all values

         tested). For this test the standardised dates were restandardised across all stations for each year.

The quality control found no impossible dates and no consecutive identical dates. Five dates (0.07%) were outside 3 sd per

140 series and ten dates (0.13%) were outside 3 sd of all series in a given year. These are very low rates, which points to a good internal consistency and likely a good quality of the data.

There are many „non-firsts" (2.47%), for which the original documents sometimes provide explanations. Since these are deviations from the observation instructions, we flagged them nevertheless. Interestingly, we found a relatively high rate of dates outside the range given by MeteoSwiss, namely 3.56%. Of these, most (60%) concerned leaf colouring of beech and

145 22.7% concerned the flowering of apple. The range given by MeteoSwiss refers to a 3 sd range per phase and altitude region. This means that for a normal distribution, 0.3% outliers are expected, however, we find ten times this amount. At the same time, only six are picked up by the other tests, which indicates that most of these outliers are consistent with the other observations both in space and time.

The QC methods test for outliers and exceptionally wide distributions, but not for the opposite, too narrow distributions. Here,

it is noteworthy that the data set has only one hazel flowering event before the start of the year, whereas we might expect this to occur more frequently.

Long series (>= 20 years) were checked for temporal inhomogeneities (caused, e.g., by a change of observed plant) following the method described in Auchmann et al. (2018). In short, 3 different statistical tests are applied to each phenological series and the agreement between the tests determines the significance of an inhomogeneity (significant when at least 2 tests agree

on an inhomogeneity). At least 3 correlated reference series are required to run the tests: this requirement limits the number of tested series to 51 (out of 56 long series). Only one series was found significantly inhomogeneous (Fig. 6).

In summary, BernClim data are expert data and subject to uncertainties. These depend on the observability of the phenomena and the speed of the development. Spring phases are typically relatively clearly defined (±1 days), whereas the autumn phases have larger uncertainties (typically ±3.5 days, see Brügger 1998).

## 4. Analyses

Figure 7 shows the day of year of hazel flowering in the BernClim network in 1971 (172 observations) and in 2017 (16 observations). The figure is supplemented with data from the Swiss Pheonological network SPN as well as from two Citizen Science projects, OpenNature and Phaenonet for 2017 (www.OpenNature.ch, www.PhaenoNet.ch). For the BernClim data, observations from the same station are joined graphically. Contiguous diamonds thus show the variation within one station across different observation sites, which may be larger than those on a regional scale. Note that flowering occurred ca. 40 days earlier in the year 2017 as compared to 1971.

Several series cover more than 40 years. As an example for a long time series, Figure 8 shows the start of blossom of the Apple tree from 9 sites at Wyssachen. The series clearly show a trend towards earlier flowering dates over the observation period. The series also shows a shift in the late 1980s. This shift is well documented in many other series (see also Reid et al. 2013). It is also found in European or even Northern Hemispheric spring snow cover (Brönnimann 2015). A change in late winter temperature around the late 1980s, albeit smaller than in observations, is also found in forced atmospheric model simulations, implying that part of this change was due to an overlap of forcing factors such as greenhouse gases, sea-surface temperatures (El Niño 1986/7, La Niña 1988/9), volcanic eruptions, and other effects (Brönnimann et al. 2006). The BernClim Data can thus help to better analyse this step-wise climatic and ecological change.

## 5. Data availability

The data presented and described in this paper are available in the data repository PANGAEA: https://doi.org/10.1594/PANGAEA.900103 (Rutishauser et al. 2019)

## 6. Conclusions

A plant phenological data set spanning almost 50 years is published as a data set on PANGEAE. Subsequently the data will be added to the PEP725 data and also available from the Geoportal of the Canton of Berne. The series were quality controlled. It should be noted that BernClim-Data are expert data and subject to uncertainties. Quality control procedures were performed to flag uncertain observations.

Although the number of stations decreased rapidly after the initial phase of the network, five long-term series remained which allow a 50-year view with almost no observer changes. Despite the sparseness, they still cover spatial variability of climatically relevant plant development stages of four species in three typical climate zones of Switzerland. Inhomogeneity tests suggest that stepwise changes are rarely driven by observational artifacts such as changes in observers, definitions or station changes, revealing a strong consistency within long time series that underline the quality of the data. In the future, the data series could be continued and merged with citizen science data and platforms such as PhaenoNet and OpenNature (Lehmann et al. 2018).

As methodologies evolved, the integration of high resolution data sets in space are more easily combinded with longterm data as the BernClim observations.

This paper only describes the phenological data. The rich (daily) winter data remain to be explored further. BernClim data may help to constrain further relevant indices such as leaf area or NDVI on a small scale. As indicated by Rutishauser et al (2007) and Stöckli et al (2008), the data have the potential to locally extend satellite data back to 1970, and they have the potential to study biological processes on the local level with continuous evidence over five decades.

**Author contribution** TR, FJ and SB conceived the study. CR, AB, PB and numerous observers collected the data. TR, YB, CP, LV, DL, LM, SB processed and analysed the data. TR and SB prepared the manuscript with contributions from all co-authors.

**Competing interests.** The authors declare that they have no conflict of interest.

**Acknowledgements.** We would like to thank all observers of BernClim who devoted a lot of time and effort into collecting data for our network. The work was supported by Swiss GCOS office (project PhenoClass) and the Swiss National Science Foundation (project 139945). The paper is dedicated to Bruno Messerli who passed away in February 2019.

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

**Tables**

**Table 1**: Complete BernClim observation programme. General flowering (also defined as full flowering) refers to the development stage when 50% of the blossoms are „open".

| Summer observations | Winter observations |
|---|---|
| Plant phenology | Snow and fog |
| hazel (*Coryllus avellane,* general flowering, pollen release) <br> dandelion (*Taraxacum officinale,* general flowering) <br> apple trees (*Pyrus malus,* general flowering) <br> beech (*Fagus sylvatica,* leave colouring) | Number of days with snow cover <br><br> Number of days with fog (visibility 0-200 m /200-1000 m), Time of fog clearing |
| Additional observations <br> date of wheat harvest (*Triticum vulgare*). <br> larch (*Larix decidua,* needle coloring), <br> coltsfoot (*Tussilago farfara,* general flowering) <br> red elder (*Sambucus racemosa,* general flowering) <br> rowan (*Sorbus aucuparia*, ripe fruits) <br> potato (*Solanum tuberosum,* planting, general flowering, the end of harvest) | |
| | Comment <br> daily, 07.00-08:00 local time |

**Table 2:** Plant specific, biological limits in days of year (DoY) with respect to five altitude ranges (MeteoSwiss, personal communication)

| Altitude | <500m asl | 500-799 m | 800-999 m | 1000-1199m | >1200 m |
|---|---|---|---|---|---|

| Phases | min | max | min | max | min | max | min | max | min | max |
|---|---|---|---|---|---|---|---|---|---|---|
| Hazel, flowering | -20 | 110 | 0 | 120 | 0 | 120 | 20 | 120 | 30 | 130 |
| Dandelion, flowering | 80 | 130 | 90 | 150 | 90 | 150 | 100 | 150 | 100 | 170 |
| Apple tree, flowering | 90 | 140 | 90 | 160 | 100 | 160 | 110 | 160 | 120 | 160 |
| Beech, leaf colouring | 250 | 310 | 250 | 310 | 240 | 310 | 240 | 300 | 230 | 300 |

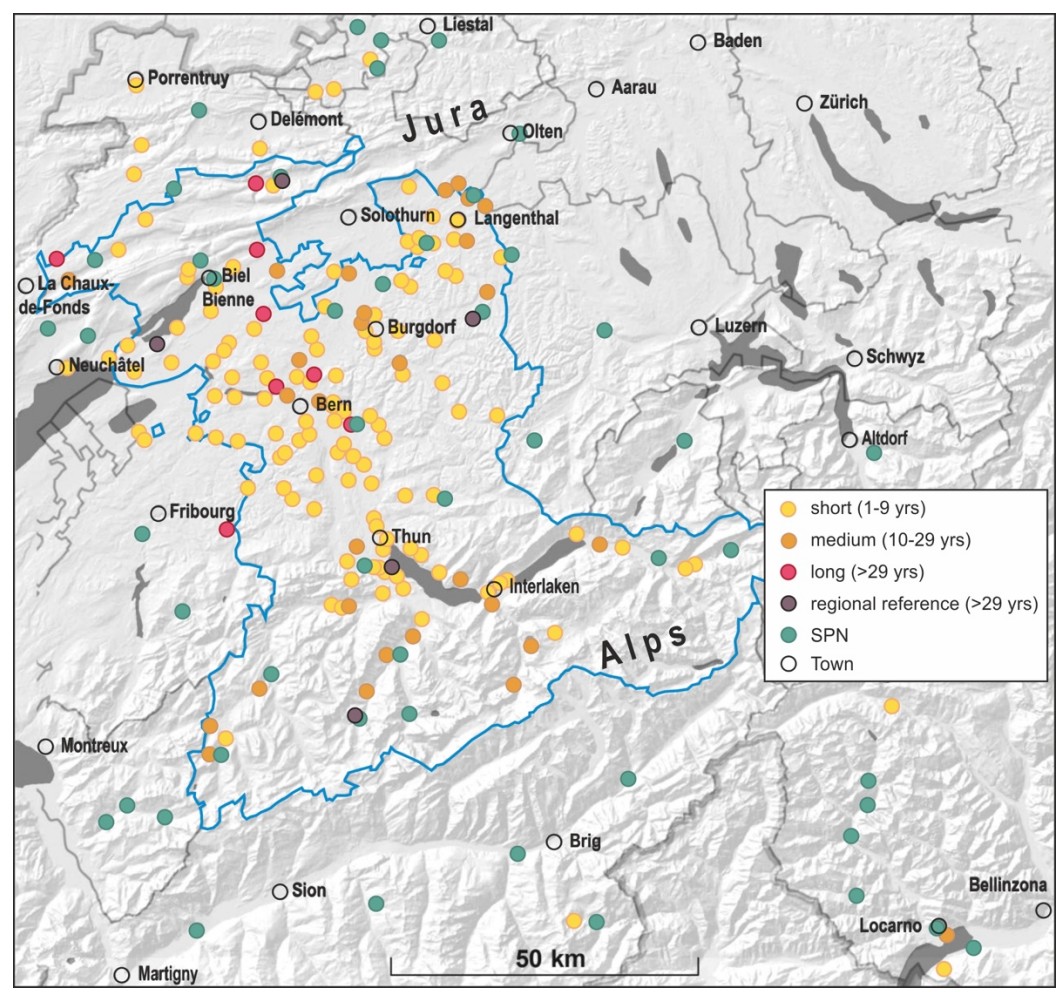

*Figure 1. Map of the BernClim stations as well as stations of the Swiss Phenological Network SPN (adapted from Jeanneret and Rutishauser 2012).*

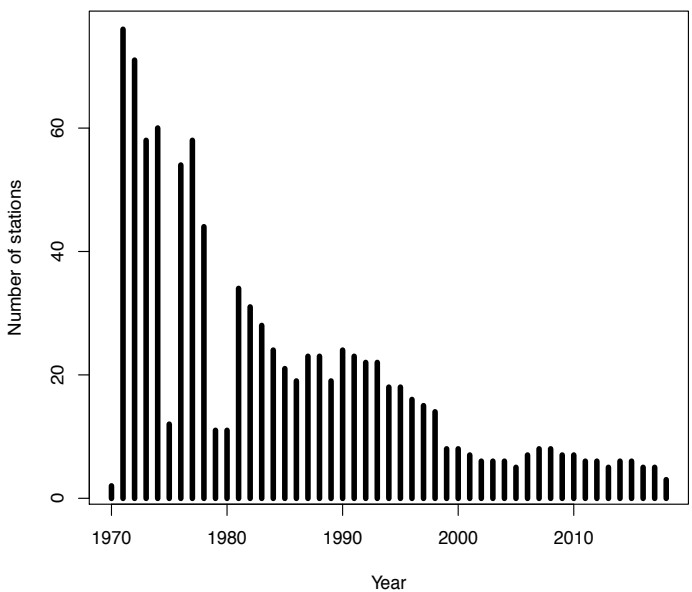

**Figure 2.** Development of the number of stations in BernClim since 1970 (updated from Jeanneret and Rutishauser 2012).

UNIVERSITAET BERN          UNIVERSITE DE BERNE
GEOGRAPHISCHES INSTITUT    INSTITUT GEOGRAPHIQUE
  Klimaforschung             Recherche climatologique

                                      Beobachtungsposten Nr.  4954.1
                                      Poste d'observation no

MELDEBLATT FUER PHAENOLOGISCHES EREIGNIS     Apfelbaum Vollblüte
FORMULAIRE POUR PHENOMENE PHENOLOGIQUE       Pommier pleine floraison     2004

| Standort Lieu | Koordinaten Coordonnées | Höhe Altitude | Exposition | Hangneigung Inclinaison | Sorte u. Bemerkungen Sorte et remarques | Datum Date |
|---|---|---|---|---|---|---|
| 1 Dorf / Koranden | 629 / 675 // 214 / 275 | 710 | flach | — | Sauergr. / Berner. / Boskop | 19.5. |
| 2 Stäublere | 629 / 575 // 214 / 175 | 720 | NE | 33% | Sauergr. / Berner. Boskop | 19.5. |
| 3 Löh | 629 / 290 / 214 / 350 | 750 | NE | ·20% | Sauergr. / Berner. Boskop Jonaton | 19.5 |
| 4 Bergli | 629 / 700 // 214 / 725 | 760 | S | 40% | Jonatan / Bernerrosen & Grakenapfel, Bona. | 17.5. |
| 5 Bödeli | 629 / 700 // 214 / 350 | 720 | S | 40% | Sauergr. / Bonapfel, Boskop Bernerrosen | 16.5. |
| 6 Ofen | 629 / 730 / 214 / 575 | 750 | WSW | 20% | Boskop / Bernerrosen Klara / Sauergrauech | 17.5 |
| 7 Lager, Garten | 629 / 825 / 214 / 200 | 740 | W | 20% | Jagreal Ontario / Goldpar mahel / Klara. Spartan | 17.5. |
| 8 Neuhauser | 629 / 830 // 214 / 100 | 740 | W | 25% | Donziga Kant / unbekannt Jerken | 17.5. |

ORIGINAL    bitte bis am 1. Dezember an      Ort und Datum
            das Institut zurücksenden        Lieu et date   Wyssachen, 19.5.04
            à retourner à l'Institut
            jusqu'au 1er décembre .          Unterschrift / Signature  A. Bernasconi

295

***Figure 3**. Example of an observation sheet for plant phenological phases (Jeanneret and Rutishauser 2012).*

UNIVERSITAET BERN          UNIVERSITE DE BERNE          Beobachtungsposten Nr.  4954.1
GEOGRAPHISCHES INSTITUT    INSTITUT GEOGRAPHIQUE        Poste d'observation no
Klimaforschung            Recherche climatologique      Winter / hiver  2003 / 04

MELDEBLATT FUER WINTERBEOBACHTUNG          S C H N E E  /  N E I G E
FORMULAIRE POUR OBSERVATION D'HIVER

Versuchsfeld       A Horizontal:    Schneebedeckung u. Schneehöhe / Couverture et hauteur de neige
Surface d'essai    B N-Exposition:  Schneebedeckung           / Couverture de neige
(ca. 10x10 m²)     C S-Exposition:  Schneebedeckung           / Couverture de neige

Signaturen    ✳ Schneebedeckung  / couverture de neige
Signes        O Kein Schnee       / pas de neige
              — Nicht beobachtet  / pas d'observation

A Schneehöhe gemessen: Sager 629 825 ‖ 214 200   flaches Stück im Garten, 740m

| | Ortsname Lieu | Koordinaten Coordonées | Höhe Altitude | Exposition | Hangneigung Inclinaison | Bemerkungen Remarques |
|---|---|---|---|---|---|---|
| A | Koronten | 629 600 ‖ 214 375 | 710 | flach | — | Talsohle mit Wiese |
| B | Stäubleren | 629 575 ‖ 214 925 | 740 | NE | 27% | Wiesen |
| C | Ofen/Oseli | 629 875 ‖ 214 475 | 770 | S | 40% | Wiesen |

| | Oktober | | | November | | | Dezember | | | Januar | | | Februar | | | März | | | April | | |
|---|---|---|---|---|---|---|---|---|---|---|---|---|---|---|---|---|---|---|---|---|---|
| | A | B | C | A | B | C | A | B | C | A | B | C | A | B | C | A | B | C | A | B | C |
| | ✳ | ✳ | ✳ | cm | ✳ | ✳ | cm | ✳ | ✳ | cm | ✳ | ✳ | cm | ✳ | ✳ | cm | ✳ | ✳ | cm | ✳ | ✳ |
| 1 | 0 | | | 0 | | | 0 | | | 6 | ✳ | ✳ | 22 | ✳ | ✳ | 10 | ✳ | ✳ | 0 | | |
| 2 | 0 | | | 0 | | | 0 | | | 12 | ✳ | ✳ | 15 | ✳ | 0 | 10 | ✳ | ✳ | 0 | | |
| 3 | 0 | | | 0 | | | 0 | | | 10 | ✳ | ✳ | 10 | ✳ | 0 | 8 | ✳ | 0 | 0 | | |
| 4 | 0 | | | 0 | | | 0 | | | 8 | ✳ | ✳ | 10 | ✳ | 0 | 6 | ✳ | 0 | 0 | | |
| 5 | 0 | | | 0 | | | 0 | | | 7 | ✳ | ✳ | 8 | ✳ | 0 | 5 | ✳ | 0 | 0 | | |
| 6 | 0 | | | 0 | | | 0 | | | 6 | ✳ | ✳ | 7 | ✳ | 0 | 3 | ✳ | 0 | 0 | | |
| 7 | 0 | | | 0 | | | 0 | | | 6 | ✳ | ✳ | 0 | ✳ | 0 | ▼ 0 | 0 | ✳ | 4 | ✳ | ✳ |
| 8 | 2 | ✳ | ✳ | 0 | | | 0 | | | 6 | ✳ | 0 | 0 | ✳ | 0 | ▼ 0 | 0 | ✳ | 0 | 0 | 0 |
| 9 | 0 | | | 0 | | | 0 | | | 4 | ✳ | 0 | 2 | ✳ | ✳ | 4 | ✳ | ✳ | 0 | | |
| 10 | 0 | | | 0 | | | 0 | | | 0 | 0 | 0 | 2 | ✳ | 0 | 0 | ✳ | 0 | 1 | ✳ | ✳ |
| 11 | 0 | | | 0 | | | 0 | | | | | | 4 | ✳ | ✳ | 5 | ✳ | 0 | 0 | 0 | 0 |
| 12 | 0 | | | 0 | | | 0 | | | | | | 5 | ✳ | ✳ | 4 | ✳ | 0 | 0 | | |
| 13 | 0 | | | 0 | | | 0 | | | | | | 3 | ✳ | 0 | 0 | ✳ | 0 | 0 | | |
| 14 | 0 | | | 0 | | | 0 | | | | | | 0 | 0 | 0 | 0 | 0 | 0 | 0 | | |
| 15 | 0 | | | 0 | | | 1 | ✳ | ✳ | 1 | ✳ | ✳ | 0 | 0 | 0 | 0 | | | 0 | | |
| 16 | 0 | | | 0 | | | 3 | ✳ | ✳ | 5 | ✳ | ✳ | 0 | 0 | ✳ | 0 | | | 0 | | |
| 17 | 0 | | | 0 | | | 2 | ✳ | 0 | 0 | 0 | 0 | 0 | 0 | ✳ | 0 | | | 0 | | |
| 18 | 0 | | | 0 | | | 2 | ✳ | 0 | 3 | ✳ | ✳ | 2 | ✳ | ✳ | 0 | | | 0 | | |
| 19 | 0 | | | 0 | | | 2 | ✳ | 0 | 8 | ✳ | ✳ | 2 | ✳ | ✳ | 0 | | | 0 | | |
| 20 | 0 | | | 0 | | | 0 | 0 | 0 | 22 | ✳ | ✳ | 2 | ✳ | ✳ | 0 | | | 0 | | |
| 21 | 0 | | | 0 | | | 0 | 0 | 0 | 12 | ✳ | ✳ | 0 | 0 | 0 | 0 | | | 0 | | |
| 22 | 0 | | | 0 | | | 6 | ✳ | ✳ | 10 | ✳ | ✳ | 0 | 0 | ✳ | 0 | | | 0 | | |
| 23 | 0 | | | 0 | | | 10 | ✳ | ✳ | 10 | ✳ | ✳ | 10 | ✳ | ✳ | 0 | | | 0 | | |
| 24 | 10 | ✳ | ✳ | 0 | | | 8 | ✳ | ✳ | 10 | ✳ | ✳ | 20 | ✳ | ✳ | 12 | ✳ | ✳ | 0 | | |
| 25 | 7 | ✳ | ✳ | 0 | | | 7 | ✳ | ✳ | 13 | ✳ | ✳ | 15 | ✳ | ✳ | 23 | ✳ | ✳ | 0 | | |
| 26 | 5 | ✳ | 0 | 0 | | | 7 | ✳ | ✳ | 15 | ✳ | ✳ | 18 | ✳ | ✳ | 15 | ✳ | ✳ | 0 | | |
| 27 | 2 | ✳ | 0 | 0 | | | 6 | ✳ | ✳ | 18 | ✳ | ✳ | 15 | ✳ | ✳ | 15 | ✳ | ✳ | 0 | | |
| 28 | 0 | ✳ | 0 | 7 | ✳ | ✳ | 3 | ✳ | 0 | 25 | ✳ | ✳ | 12 | ✳ | ✳ | 13 | ✳ | ✳ | 0 | | |
| 29 | 0 | ✳ | 0 | 3 | ✳ | ✳ | 2 | ✳ | ✳ | 32 | ✳ | ✳ | 10 | ✳ | ✳ | 8 | ✳ | 0 | 0 | | |
| 30 | 0 | 0 | 0 | 1 | ✳ | 0 | 2 | ✳ | 0 | 35 | ✳ | ✳ | | | | 0 | ✳ | 0 | 0 | | |
| 31 | 0 | | | | | | 3 | ✳ | ✳ | 30 | ✳ | ✳ | | | | 0 | 0 | 0 | | | |
| TOTAL | 5 | 7 | 3 | 3 | 3 | 2 | 15 | 15 | 10 | 25 | 25 | 23 | 20 | 29 | 13 | 15 | 20 | 10 | 2 | 2 | 2 |

Ergebnisse:  Total der Tage mit Schneebedeckung / Jours avec couverture de neige (0 – λ)    ▼ angeschneit

A: 85  Tage / jours   B: 101  Tage / jours   C: 63  Tage / jours

Einsendetermin: 1. Juni / juin          Beobachter / observateur: A. Bernasconi

*Figure 4. Example of an observation sheet for snow (Jeanneret and Rutishauser 2012).*

MELDEBLATT FUER WINTERBEOBACHTUNG:    N E B E L / B R O U I L L A R D
FORMULAIRE POUR OBSERVATION D'HIVER:

| Zu beobachten | A | Nebel zwischen o7.00 und o8.00 / brouillard entre o7.00 et o8.00 |
| à observer | B | Zeit der Nebelauflösung / temps de la dissipation du brouillard |

| Signaturen | 2 | Nebel (Sichtweite o- 2oo m) / brouillard (visibilité o- 2oo m) |
| signes | 1 | Nebel (Sichtweite 2oo-1ooo m) / brouillard (visibilité 2oo-1ooo m) |
| | o | Kein Nebel / pas de brouillard |
| | — | Nicht beobachtet / pas d'observation |

| | Ortsname Lieu | Koordinaten Coordonnées | Höhe Altitude | Exposition | Hangneigung Inclinaison | Bemerkungen Remarques |
|---|---|---|---|---|---|---|
| A+B | Sager | 629/825 ‖ 214/2oo | 740 m | W | 20% | offene Hanglage |

| | September | | Oktober | | November | | Dezember | | Januar | | Februar | | März | |
|---|---|---|---|---|---|---|---|---|---|---|---|---|---|---|
| | A | B | A | B | A | B | A | B | A | B | A | B | A | B |
| 1 | o | | o | | o | | o | | o | | o | | o | |
| 2 | o | | o | | o | | o | | o | Hoch≡ bleib | o | | o | |
| 3 | o | | o | | o | | o | | o | Hoch≡, bleibt | o | | o | |
| 4 | o | | o | | o | | ≡1 | ab 10·00 Hoch≡. | o | | o | | o | |
| 5 | o | | o | | o | Hoch≡ ab 17.00 | o | | o | | o | | o | |
| 6 | o | | o | | o | | ≡1 | 10.00 | o | | o | | o | |
| 7 | o | | o | | o | Hoch≡ bis 9.05 | o | | o | | o | | o | |
| 8 | o | | o | | o | Hoch≡ bis 10.00 | o | | ≡2 | 9.00 | o | | o | |
| 9 | o | | o | | o | | ≡1 | bleibt | o | | o | | o | |
| 1o | o | | o | | o | | o | | o | | o | | o | |
| 11 | o | | o | Hoch≡ bis 10.00 | ≡2 | 13.00 | o | | o | | o | | o | |
| 12 | o | | ≡1 | ab 10.00 Hoch≡ | o | | o | | o | | o | Hoch≡ bis 10.00 | o | |
| 13 | o | | o | | o | | o | | o | | o | | o | |
| 14 | o | | ≡1 | ab 11.00 Hoch≡ | o | | o | | o | | o | Hoch≡ bis | o | |
| 15 | o | | o | Hoch≡ bis 10.00 | o | | o | | o | | o | ab 13.00 Hoch≡ | o | |
| 16 | o | | o | Hoch≡ bis 10.00 | o | | o | | o | | o | Hoch≡ bis 10.00 | o | |
| 17 | o | | o | Hoch≡ bis | o | | o | | o | | o | Hoch≡ bleibt | o | |
| 18 | o | | ≡1 | ab 10.00 Hoch≡ bis 13.00 | o | | o | | o | | o | | o | |
| 19 | o | | ≡2 | 12.00 | o | | o | | o | | o | Hoch≡ bis 10.00 | o | |
| 2o | o | | o | | o | | o | | o | | o | | o | |
| 21 | o | | o | | o | | o | | o | | o | | o | |
| 22 | o | | o | | o | | o | | o | Vormittags | o | | o | |
| 23 | o | | o | | o | | o | | o | | o | | o | |
| 24 | o | | o | | o | | o | | o | | o | | o | |
| 25 | o | | o | | o | | o | | o | | o | | o | |
| 26 | o | | o | | o | | o | | o | | o | | o | Hoch≡ bis 19.00 |
| 27 | o | | o | Hoch≡ bis 10.00 | o | | o | | o | | o | | o | |
| 28 | o | | o | | o | | o | | o | | o | | o | |
| 29 | o | | o | | o | ab 10.00 Hoch≡ | o | | o | | | | o | |
| 3o | o | Hoch≡ bis 17.00 | o | | o | | o | Hoch≡ bleibt | o | | | | o | |
| 31 | | | o | | | | o | | o | | | | o | |
| TOTAL | o | | 4 | | 1 | | 3 | | 1 | | 0 | | 0 | |

Zahl der Nebeltage (Oktober bis März o - 1ooo m)
Jours de brouillard (octobre jusqu'à mars, visibilité o - 1ooo m)    9

Erster Frost (fakultativ) - Datum:    8. 10. 03, kombiniert mit dem ersten ✱
Première gelée (facultativ) - date:

| Einsendetermin: 1.Juni / juin |

Beobachter / observateur
A. Bernasconi

Figure 5. Example of an observation sheet for fog (Jeanneret and Rutishauser 2012).

300

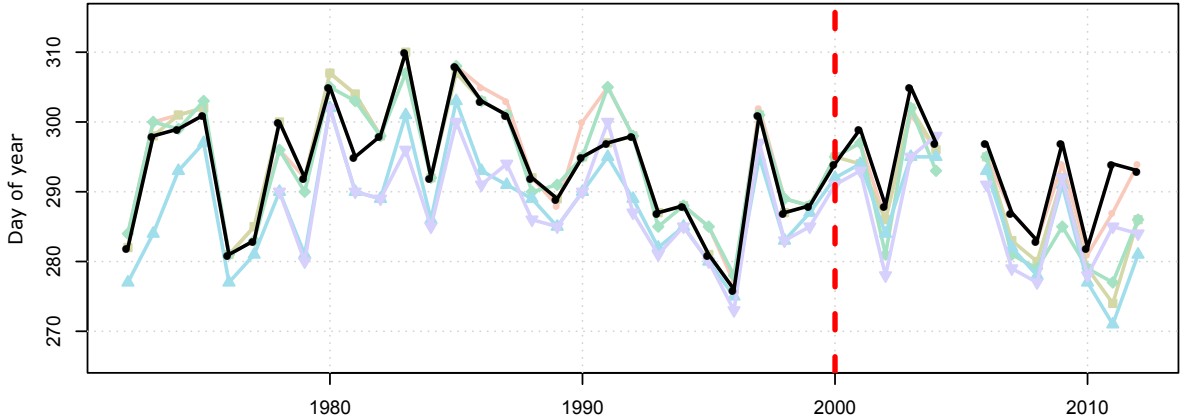

***Figure 6.*** *Inhomogeneous series for the leaf colouring of the beech in Wyssachen- Oeseliwaeldli (bold black line). The coloured lines represent 5 other series of the leaf colouring of beech (reference series) in other parts of the community of Wyssachen. The leaf colouring in the inhomogeneous series occurs on average about 3 days later than expected after the year 2000 (red vertical line).*

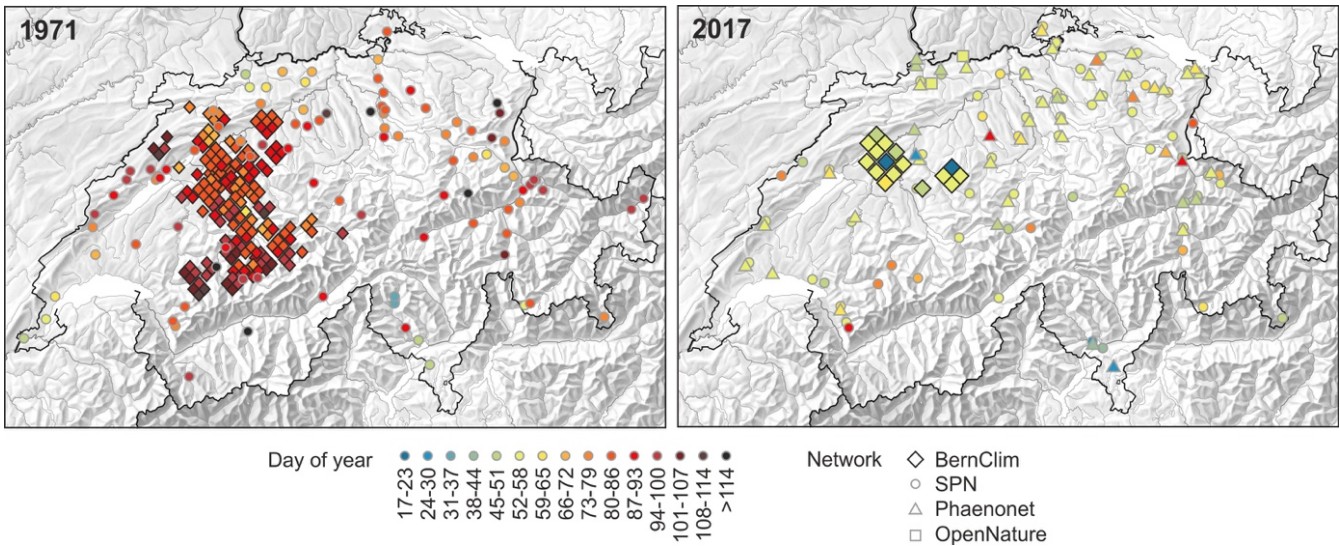

***Figure 7.*** *General flowering of hazel in BernClim (diamonds) and SPN (circle) data in 1971 (left) and 2017 (right). The right figure also shows data from two Citizen Science Projects PhaenoNet (triangle) and OpenNature (squares) (updated from Lehmann et al. 2018).*

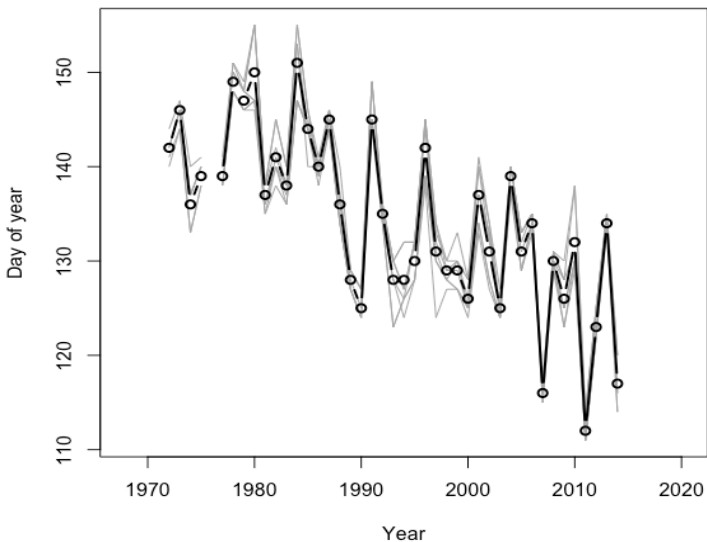

***Figure 8.*** *BernClim apple flowering dates from nine sites (grey lines) of station Wyssachen (710 to 760 m a.s.l.) Black lines and circle denote station mean dates.*