# Peer review of "The BernClim plant phenological data set from the Canton of Bern (Switzerland) 1970–2018"

_Earth System Science Data, 2019_

## Referee Comment (RC1) · Anonymous Referee #1 · 22 Jul 2019

General comments

In their Data Description paper, 'The BernClim plant phenological data set from the Canton of Bern (Switzerland) 1970–2018' Rutishauser and colleagues describe long-term phenological data from three tree species and one herb collected since 1970 (and continuing today). The data are relatively unique in having the same observers at most sites over the (long) observation period; and thus provide an important comparison for data collected by varied observers over time (which is far more common, based on my experience). The authors review some basics of the data collection, quality control on the data and provide some simple but very nice visuals of the data, including some

basic information on how they capture extending growing seasons with climate change. These data add to one of the better areas of the world for high-quality long-term data and are important for understanding climatic control on plant phenology in the past and what it means for plants in a future, hotter planet.

Specific comments

This paper is generally very well written and easy to read, but I was confused about a few things that I think minor re-structuring could address.

(1) Given that non-first dates were flagged in quality control I think the data collection must have been focused on first events. However this was not clearly stated (at least not clearly enough for me). If the data are focused on first events, please state it more clearly. If not it would be helpful to know why non-first dates were flagged. (2) I would have appreciated a little info on who the observers were and how they were found and enlisted in the project. (3) Somewhat related to (2) I would move up the Jeanneret & Rutishauser 2012 ... For example, an overview of what you will cover in the 'observation network and data' could be followed by 'more details, such as on how observers were located, trained and details on [insert a few more important details covered in other paper] are given in Jeanneret & Rutishauser 2012. (4) I additionally wondered if the trees were in forests, cultivated systems, clonal gardens or what?

Technical comments - If possible with the journal's style guidelines it would be nice to see the species names italicized in the abstract. - I would have liked a quick explanation of what regime shift was referenced in the abstract. For example, instead of 'the regime shift in the late 1980s,' it could read 'a regime shift in the late 1980s observed across numerous other phenological and meteorological datasets.' - Given the focus on 'first dates' I would reword line 74 "First observations were performed in 1970' to 'Observations began in 1970.' - Exposition is often called 'aspect' in my world, I might mention this once in the abstract and once in the main text: exposition (aspect) .... - Line 103: 123 500 data collected over what period of years? The number isn't that

meaningful without a time-window. - Robert Brugger is an expert in what? - Line 106: extra period at end of sentence - day of year is in inconsistently spelled (day-of-year, Days of Year etc.) throughout manuscript; whichever seems find to me so long as it is consistent. - Line 137-8 and 160 - 'the beech' and 'the Apple tree' sounds a bit odd, it should be apple, not Apple I believe and I think 'the' may not be needed. I suggest instead "...leaf colouring of beech and 22.7% concerned flowering of apple." - Figure 6: The colored versus black lines are not well explained (I assume black means something different? As 5 colored lines are mentioned in the caption and the black line is a 6th line, I believe), nor is there information on the red dashed line. I would also mention the data gap in the figure. - Figure 7: For consistency I think it should be 'hazel' and not 'hazelnut.'

---

## Referee Comment (RC2) · Anonymous Referee #2 · 18 Aug 2019

Review ESSD-2019-101, Bern phenology data set

Thanks to authors for removing data access barriers. Data now downloads easily, looks very clean. Confirm 7414 data records.

Page 2 line 56: "suggest that the data different data sources are complementary" Something wrong with text here. You mean 'data from different sources' or instead 'the different data sources'?

Page 4 line 109 (and Figures 4 and 5): daily data for fog and snow using these forms but those data not included in this data product? A bit confusing to read about daily winter data while not knowing how or where archived and how accessed. Authors do not need to show those data, but if they take the trouble to mention the daily observations and to show the forms, readers should learn at least how to access those data?

Page 4 line 121: "Each series was standardized" What does 'standardized mean in this context? This refers to the assembly of stations by coordinates into a shared zip code? Or this means that each series underwent the 6 QC steps listed soon after? Some clarification, s.v.p.

Page 5 line 130: Again this word "standardized" now referring to DoY values. "standardized dates were restandardized". What does this mean?

Page 6 line 178 to 182, discussion of future monitoring. Note the word "could' on line 178! As observers (and, in some cases, trees) age out of the survey, will this record cease? Figure 7 suggests that those other networks will not retain the high spatial resolution of the Bern data? Line 179: here reader finds that winter data awaits analysis. That data not otherwise available? Analysis by whom, and when expected? Line 180: how would 50% blossom data prove relevant to leaf area or NDVI? Do the authors have specific examples? Not clear to this reviewer?

Figure 1: even though I know the region, the various subtle greyscale and sizes of dots on a grey background confuse this reader. Get the figure in colour, or change some symbol shapes (diamonds, stars) to better distinguish station types? Need a bold outline of canton Bern! If no page charges, and therefore no colour penalty, why not use a color background? Many exist.

Figure 2: this reader admires observer persistence and the long continuity of these records. But at some point the data become so few that they cease to provide a valid spatial representation. At what point? I believe others have addressed this question of minimum spatial requirements?

Figure 6: Presumably the bold black line represents these data? If so, designate in the figure legend. The inhomogeneity emerges when these data begin to show consistent later DoY date than all other five reference sites? The authors do not share nor propose an explanation for this inhomogeneity? Because I find location Wyssachen many times in the common beech data subset, I can not tell whether the authors flagged and removed this particular time series or retained it?

Figure 7: the colour scale shown with reference to circle data also applies to diamonds? If so, this figure confirms the 40-day advancement mentioned on page 5, line 158?

---

## Author Response (AR1)

Dear referees, dear editor

In the name of all co-authors I would like to thank you for the careful handling of the manuscript and the number of constructive comments that substantially increased the quality of our manuscript. Please find below our replies to the questions and suggestions raised. We are glad to include all changes in the revised version of the manuscript.

Best regards, This Rutishauser

**Anonymous Referee #1**

General comments

In their Data Description paper, 'The BernClim plant phenological data set from the Canton of Bern (Switzerland) 1970–2018' Rutishauser and colleagues describe long- term phenological data from three tree species and one herb collected since 1970 (and continuing today). The data are relatively unique in having the same observers at most sites over the (long) observation period; and thus provide an important comparison for data collected by varied observers over time (which is far more common, based on my experience). The authors review some basics of the data collection, quality control on the data and provide some simple but very nice visuals of the data, including some basic information on how they capture extending growing seasons with climate change. These data add to one of the better areas of the world for high-quality long-term data and are important for understanding climatic control on plant phenology in the past and what it means for plants in a future, hotter planet.

Specific comments

This paper is generally very well written and easy to read, but I was confused about a few things that I think minor re-structuring could address.

We appreciate the general comments of the reviewer and thankfully address specific comments and suggestions below.

(1) Given that non-first dates were flagged in quality control I think the data collection must have been focused on first events. However this was not clearly stated (at least not clearly enough for me). If the data are focused on first events, please state it more clearly. If not it would be helpful to know why non-first dates were flagged.
We assume that there exists some confusion in the terminology. "First date observations" can be defined as the first flowering/budburst of a plant. This would relate to the 1% of a phenological event. In our dataset, we find observations that recorded the date when 50% of the flowers/buds opened. Thus, non-first events were flagged when they relate to a repeated observation of 50%.

(2) I would have appreciated a little info on who the observers were and how they were found and enlisted in the project.
We added the following sentence "Overall, more than 200 volunteers were recruited for observing in 1971 through the teacher training program of the Institute of Geography. A large number of observers have a training in public school teaching or family doctors, and have a strong, intrinsic motivation for observing natural phenomena and processes."

(3) Somewhat related to (2) I would move up the Jeanneret & Rutishauser 2012 ... For example, an overview of what you will cover in the 'observation network and data' could be followed by

'more details, such as on how observers were located, trained and details on [insert a few more important details covered in other paper] are given in Jeanneret & Rutishauser 2012.
The reference is moved and additional information added in the text as suggested

(4) I additionally wondered if the trees were in forests, cultivated systems, clonal gardens or what?
Added in the text. Most sites and the plants were observed in cultivated systems. No clonal individuals were observed

Technical comments - If possible with the journal's style guidelines it would be nice to see the species names italicized in the abstract. done

- I would have liked a quick explanation of what regime shift was referenced in the abstract. For example, instead of 'the regime shift in the late 1980s,' it could read 'a regime shift in the late 1980s observed across numerous other phenological and meteorological datasets.' Done

- Given the focus on 'first dates' I would reword line 74 "First observations were performed in 1970' to 'Observations began in 1970.' done

- Exposition is often called 'aspect' in my world, I might mention this once in the abstract and once in the main text: exposition (aspect) done

- Line 103: 123 500 data collected over what period of years? The number isn't that meaningful without a time-window. We include "During the intensive initial phase of the network"

- Robert Brugger is an expert in what?
We added to the text "... an expert in biology, plant physiology and phenology (Robert Brügger)"

- Line 106: extra period at end of sentence done

- day of year is in inconsistently spelled (day-of-year, Days of Year etc.) throughout manuscript; whichever seems find to me so long as it is consistent.
We checked the complete manuscript and used day of year (DoY)

- Line 137-8 and 160 - 'the beech' and 'the Apple tree' sounds a bit odd, it should be apple, not Apple I believe and I think 'the' may not be needed. I suggest instead "...leaf colouring of beech and 22.7% concerned flowering of apple."
done

- Figure 6: The colored versus black lines are not well explained (I assume black means some-thing different? As 5 colored lines are mentioned in the caption and the black line is a 6th line, I believe), nor is there information on the red dashed line. I would also mention the data gap in the figure.
done

- Figure 7: For consistency I think it should be 'hazel' and not 'hazelnut.' done

**Anonymous Referee #2**

Good product. See small list of corrections and questions in supplement.

Please also note the supplement to this comment: https://www.earth-syst-sci-data-discuss.net/essd-2019-101/essd-2019-101-RC2- supplement.pdf

Review ESSD-2019-101, Bern phenology data set

Thanks to authors for removing data access barriers. Data now downloads easily, looks very clean. Confirm 7414 data records.

Page 2 line 56: "suggest that the data different data sources are complementary" Something wrong with text here. You mean 'data from different sources' or instead 'the different data sources'? corrected

Page 4 line 109 (and Figures 4 and 5): daily data for fog and snow using these forms but those data not included in this data product? A bit confusing to read about daily winter data while not knowing how or where archived and how accessed. Authors do not need to show those data, but if they take the trouble to mention the daily observations and to show the forms, readers should learn at least how to access those data?
We rephrased the section to "All original observation sheets of plant, snow and fog observations are archived at the University of Bern. During ongoing data rescue a large fraction has been photographed. To date, only plant observations have been digitized and controlled for publication."

Page 4 line 121: "Each series was standardized" What does 'standardized mean in this context? This refers to the assembly of stations by coordinates into a shared zip code? Or this means that each series underwent the 6 QC steps listed soon after? Some clarification, s.v.p.
Thanks for the remark. We replaced "Each series was standardised." with "The first four tests use absolute dates, test five is based on standardised series while for test six, for a given year, the standardised dates from all series were re-standardized." Standartisation always refers to the time series at an observation site and does not mean the assemblage of several site series into one combined series.

Page 5 line 130: Again this word "standardized" now referring to DoY values. "standardized dates were restandardized". What does this mean?
We replaced "restandardized" with "scaled" to clarify the retransformation to days of year.

Page 6 line 178 to 182, discussion of future monitoring. Note the word "could' on line 178! As observers (and, in some cases, trees) age out of the survey, will this record cease? Figure 7 suggests that those other networks will not retain the high spatial resolution of the Bern data?

We added more information about the question raised in the manuscript. "In the future, the data series could be continued and merged with citizen science data and platforms such as PhaenoNet and OpenNature (Lehmann et al. 2018). As methodologies evolved, the integration of high resolution data sets in space are more easily combined with longterm data as the BernClim observations."

Line 179: here reader finds that winter data awaits analysis. That data not otherwise available? Analysis by whom, and when expected?

There are no specific plans nor deadlines foreseen at the moment. Please see the answer to Page 4 line 109 above for more details.

Line 180: how would 50% blossom data prove relevant to leaf area or NDVI? Do the authors have specific examples? Not clear to this reviewer?

We added the following changes to the revised manuscript. "As indicated by Rutishauser et al (2007) and Stöckli et al (2008), the data have the potential to locally extend satellite data back to 1970, and they have the potential to study biological processes on the local level with continuous evidence over five decades."

Figure 1: even though I know the region, the various subtle greyscale and sizes of dots on a grey background confuse this reader. Get the figure in colour, or change some symbol shapes (diamonds, stars) to better distinguish station types? Need a bold outline of canton Bern! If no page charges, and therefore no colour penalty, why not use a color background? Many exist.

We adapted Figure 1 in the revised manuscript.

Figure 2: this reader admires observer persistence and the long continuity of these records. But at some point the data become so few that they cease to provide a valid spatial representation. At what point? I believe others have addressed this question of minimum spatial requirements?

Thank you for the remark. Minimum spatial requirements with respect to spatial representation have been addressed for many variables. We have decided not to perform quantitative analyses and refer to the in-depth analyses by Güsewell et al (2018) for the Swiss Phenology Network SPN. They find that phenological variation across Switzerland is determined by altitude, large-scale spatial trends and local deviations (e.g. due to variation among individual plants and observation error), whereas small-scale spatial dependence (correlation of neighbouring stations) is weak. Güsewell et al state that the number of stations currently included in the SPN is sufficient for precise estimates of mean onset dates of each phenophase, of long-term trends and of responses to temperature for the entire country and for three altitudinal layers. Finally, Güsewell et al. conclude that the quality of the responses of plant phenology to climatic factors and to predict changes associated with future climate warming depend critically on the quality of the underlying data.

Güsewell S, Pietragalla B, Gehrig R and Furrer R: 2018, Representativeness of stations and reliability of data in the Swiss Phenology Network, *Technical Report MeteoSwiss*, 267, 100 pp. https://www.meteoschweiz.admin.ch/home/service-undpublikationen/publikationen.subpage.html/de/data/publications/2018/6/representativeness-of-stations-and-reliability-of-data-in-the-swiss-phenology-network.html

Figure 6: Presumably the bold black line represents these data? If so, designate in the figure legend. The inhomogeneity emerges when these data begin to show consistent later DoY date than all other five reference sites? The authors do not share nor propose an explanation for this inhomogeneity? Because I find location Wyssachen many times in the common beech data subset, I can not tell whether the authors flagged and removed this particular time series or retained it?

We adapted the figure caption as the reviewer suggests. Figure 6 is shown to illustrate the inhomogeneity testing in the present study. We have not analysed the case studies with respect to explanations for each inhomogeneity arising.

Figure 7: the colour scale shown with reference to circle data also applies to diamonds? If so, this figure confirms the 40-day advancement mentioned on page 5, line 158?

Yes. Colour scale is now moved within the figure to reduce confusions. As pointed out, this figure confirms a shift of more than one month for hazel flowering within the past 50-year period. Note that the flowering of hazel is very sensitive to temperature forcing . Thus, the change shown in the figure not only shows longterm temperature changes but also interannual variability.

[revised manuscript text omitted]
 date of wheat harvest (*Triticum vulgare*). larch (*Larix decidua,* needle coloring), coltsfoot (*Tussilago farfara,* general flowering) red elder (*Sambucus racemosa,* general flowering) rowan (*Sorbus aucuparia*, ripe fruits) potato (*Solanum tuberosum,* planting, general flowering, the end of harvest) | |
| | Comment daily, 07.00-08:00 local time |

**Table 2:** Plant specific, biological limits in days of year (DoY) with respect to five altitude ranges (MeteoSwiss, personal communication)

| Altitude | <500m asl | 500-799 m | 800-999 m | 1000-1199m | >1200 m |
|---|---|---|---|---|---|

hat gelöscht: Days

hat gelöscht: Year

| Phases | min | max | min | max | min | max | min | max | min | max |
|---|---|---|---|---|---|---|---|---|---|---|
| Hazel, flowering | -20 | 110 | 0 | 120 | 0 | 120 | 20 | 120 | 30 | 130 |
| Dandelion, flowering | 80 | 130 | 90 | 150 | 90 | 150 | 100 | 150 | 100 | 170 |
| Apple tree, flowering | 90 | 140 | 90 | 160 | 100 | 160 | 110 | 160 | 120 | 160 |
| Beech, leaf colouring | 250 | 310 | 250 | 310 | 240 | 310 | 240 | 300 | 230 | 300 |

[Figure]

[Figure]

*Figure 1. Map of the BernClim stations as well as stations of the Swiss Phenological Network SPN (adapted from Jeanneret and Rutishauser 2012).*

330

[Figure]

**Figure 2.** Development of the number of stations in BernClim since 1970 (updated from Jeanneret and Rutishauser 2012).

| UNIVERSITAET BERN | UNIVERSITE DE BERNE | | | | |
|---|---|---|---|---|---|
| GEOGRAPHISCHES INSTITUT | INSTITUT GEOGRAPHIQUE | | | | |
| Klimaforschung | Recherche climatologique | | | Beobachtungsposten Nr. 4954.1 | |
| | | | | Poste d'observation no | |

MELDEBLATT FUER PHAENOLOGISCHES EREIGNIS
FORMULAIRE POUR PHENOMENE PHENOLOGIQUE

Apfelbaum Vollblüte
Pommier pleine floraison    2004

| Standort Lieu | Koordinaten Coordonnées | Höhe Altitude | Exposition | Hangneigung Inclinaison | Sorte u. Bemerkungen Sorte et remarques | Datum Date |
|---|---|---|---|---|---|---|
| 1 Dorf / Koranden | 629 675 214 275 | 710 | flach | — | Sauergr. / Berner. Boskop | 19.5. |
| 2 Häublesen | 629 575 214 175 | 720 | NE | 33% | Sauergr. / Berner. Boskop | 19.5. |
| 3 Löh | 629 290 214 350 | 750 | NE | ·20% | Sauergr. / Berner. Boskop Jonathan | 19.5 |
| 4 Bergli | 629 700 214 725 | 760 | S | 40% | Jonathan / Berner. Rosen & Gravenstein, Bona. | 17.5. |
| 5 Bödeli | 629 700 214 350 | 720 | S | 40% | Sauergr. / Bond. Apfel, Boskop Berner. Rosen | 16.5. |
| 6 Olen | 629 730 214 575 | 750 | WSW | 20% | Boskop / Berner. Rosen Klar. / Sauer. Gravenst. | 17.5. |
| 7 Lager, Garten | 629 815 214 200 | 740 | W | 20% | Jagreal Gravenst. / Gravenst. Maren / Klard. Spartan | 17.5. |
| 8 Neuhauser | 629 830 214 100 | 740 | W | 25% | Somerg. Klar. / Unbekannt Jonathan | 17.5. |

ORIGINAL    bitte bis am 1. Dezember an
das Institut zurücksenden
à retourner à l'Institut
jusqu'au 1er décembre    .

Ort und Datum
Lieu et date  *Wyssachen, 19.5.04*

Unterschrift / Signature  *A. Bernasconi*

335

***Figure 3**. Example of an observation sheet for plant phenological phases (Jeanneret and Rutishauser 2012).*

*Figure 4*. Example of an observation sheet for snow (Jeanneret and Rutishauser 2012).

Figure 5. Example of an observation sheet for fog (Jeanneret and Rutishauser 2012).

[Figure]

*Figure 6.* Inhomogeneous series for the leaf colouring of the beech in Wyssachen- Oeseliwaeldli _(bold black line)_. The coloured lines represent 5 other series of the leaf colouring of beech (reference series) in other parts of the community of Wyssachen. The leaf colouring in the inhomogeneous series occurs on average about 3 days later than expected after the year 2000 _(red vertical line)_.

**hat gelöscht:** _the_

[Figure]

*Figure 7.* General flowering of _hazel_ in BernClim (diamonds) and SPN (circle) data in 1971 (left) and 2017 (right). The right figure also shows data from two Citizen Science Projects PhaenoNet (triangle) and OpenNature (squares) (updated from Lehmann et al. 2018).

**hat gelöscht:** _hazelnut_

[Figure]

355 **Figure 8.** *BernClim apple flowering dates from nine sites (grey lines) of station Wyssachen (710 to 760 m a.s.l.) Black lines and circle denote station mean dates.*